# MAPK15 Prevents *IFNB1* Expression by Suppressing Oxidative Stress-Dependent Activation of the JNK-JUN Pathway

**DOI:** 10.3390/ijms26115148

**Published:** 2025-05-27

**Authors:** Monia Taranta, Sara Panepinto, Federico Galvagni, Lorenzo Franci, Mario Chiariello

**Affiliations:** 1Istituto di Fisiologia Clinica (IFC), Consiglio Nazionale delle Ricerche (CNR), 53100 Siena, Italy; monia.taranta@cnr.it (M.T.); lorenzofranci@cnr.it (L.F.); 2Core Research Laboratory (CRL), Istituto per lo Studio la Prevenzione e la Rete Oncologica (ISPRO), 53100 Siena, Italy; sarapanepinto92@gmail.com; 3Department of Biotechnology, Chemistry and Pharmacy, University of Siena, 53100 Siena, Italy; federico.galvagni@unisi.it

**Keywords:** MAP kinases, interferons, inflammation, signal transduction

## Abstract

Human type I interferons are crucial regulators of immune responses, essential for controlling infections and activating immune cells. Among them, Interferon Beta (IFNB1) plays a key role in inflammation, and its dysregulation is linked to various diseases, driving efforts to understand the molecular events governing its expression. Here, we identified Mitogen-Activated Protein Kinase 15 (MAPK15) as a novel regulator of *IFNB1*. Using luciferase reporter assays, gene expression analysis and Enzyme-Linked Immunosorbent Assay (ELISA), we found that *MAPK15* downregulation enhanced *IFNB1* and Interferon-Stimulated Genes expression and increased IFNB1 secretion. To unveil the underlying mechanisms, we investigated the transcription factors acting on the *IFNB1* promoter, revealing that *MAPK15* downregulation induced JUN activation. Importantly, pharmacological inhibition of c-Jun N-terminal Kinases (JNKs) supported a key role for this enzyme in JUN activation and consequent *IFNB1* expression. Ultimately, by using the antioxidant N-acetylcysteine ethyl ester (NACET), we demonstrated that oxidative stress, induced by *MAPK15* downregulation, was responsible for JUN activation and *IFNB1* expression. Overall, our findings unveil a novel mechanism by which MAPK15 modulates *IFNB1* expression, positioning this kinase as a pivotal regulator of this gene. This insight opens promising avenues for therapeutic intervention, as targeting MAPK15 activity could offer a strategy to rebalance cytokine expression in chronic inflammatory diseases characterized by immune dysregulation.

## 1. Introduction

Human type I interferons (IFNs) are a family of cytokines controlling the modulation of the immune system. They are composed of multiple genes, including 13 partially homologous IFNA (IFN-alpha) subtypes, a single IFNB1 (IFN-beta) and several less characterized single gene products [1,2]. They perform many functions to hinder the spread of infectious agents, such as viral or bacterial pathogens, stimulate immune system cells activity, and induce inflammatory pathways [3]. Indeed, type I IFNs are promptly secreted in large amounts during infections, to induce an early immune response and prime adaptive immunity. Nonetheless, it is now well established that IFNs are also constitutively expressed, in small quantities, in the absence of infection [4,5], to keep the immune system in a sort of “ready to go” state ensuring quick responses, when necessary [6]. On the other hand, dysregulation of IFNs basal levels may have detrimental effects, enhancing inflammatory responses that increase the risk of developing autoimmune diseases [5,6] and could also predispose to cancer [7]. For these reasons, understanding the molecular events that impact type I IFNs expression is highly desirable. Mitogen activated protein kinases (MAPKs or MAP kinases) are signaling proteins that efficiently transduce and amplify extracellular and intracellular stimuli, inducing responses that affect several cellular functions, including gene expression, proliferation, metabolism, motility, survival, apoptosis, and differentiation [8]. Given the central role of MAPK signaling pathways in numerous cellular processes, these kinases have emerged as attractive targets for therapeutic intervention in diseases such as cancer, autoimmune disorders, and inflammatory conditions. Recent studies have further supported this view, demonstrating that MAPK proteins represent promising candidates for cancer therapy [9], thereby reinforcing their potential as therapeutic targets. MAPK15 is the most recently identified member of the MAPK family [10] and, consequently, many aspects of its biological role remain to be characterized. However, even if the picture is still not complete, it is now evident its involvement in the management of stressful stimuli, for the maintenance of different physiological cellular functions. Specifically, MAPK15 has been implicated in controlling the DNA damage response [11,12], the protection of genomic stability by controlling PCNA levels [13] and telomerase activity [14], the regulation of autophagy [15,16] and reactive oxygen species (ROS) generation [12]. More recently, MAPK15 has been involved in ROS-induced cellular senescence thanks to its ability to control mitophagy [17] and induce NRF2-dependent antioxidant responses [18]. Interestingly, MAPK15 has been also linked to the expression of several cytokines that are part of a senescent-associated secretory phenotype (SASP) and involved in immunoregulatory and inflammatory processes [17].

Here, we explored a role for MAPK15 in controlling IFNs expression, with the aim of evaluating its potential as a therapeutic target for modulating inflammatory disorders. In particular, among the different subtypes of type I IFNs, we focused our attention on *IFNB1* because of its role as a master regulator of all type I IFNs [5,19,20]. To this aim, we decided to use the bronchial epithelial cell line 16HBE14o- since the bronchial epithelium, which expresses high levels of endogenous MAPK15 protein, is continuously exposed to external agents, representing a primary barrier for host defense, regulating both innate and adaptive immunity through the release of cytokines and type I IFNs [21,22].

## 2. Results

### 2.1. Loss of MAPK15 Results in Increased IFNB1 Response

To investigate a potential role for MAPK15 in affecting cellular levels of IFNs, we first evaluated its ability to control transactivation of the *IFNB1* promoter by performing luciferase assays. For this purpose, we employed a plasmid in which the reporter gene is under the control of *IFNB1* human promoter, which contains the four regulatory domains (PRD I-PRD IV) for the transcription factors NF-kB, AP-1 and IRF3/IRF7, as depicted in Appendix A [23]. In particular, when we used specific siRNA [12,17,24] to knockdown endogenous *MAPK15* expression, whose efficacy was verified by both Reverse Transcription quantitative Polymerase Chain Reaction (RT-qPCR) and Western blot analysis (Appendix A), luciferase experiments revealed that the reporter gene activity was significantly increased (Figure 1A). Accordingly, the analysis of *IFNB1* gene expression by RT-qPCR also demonstrated a significant increase in endogenous *IFNB1* expression in cells interfered for MAPK15, compared to controls (Figure 1B). Consistently with these results, MAPK15 overexpression significantly reduced transactivation of the *IFNB1* promoter (Figure 1C), overall suggesting a connection between MAPK15 and *IFNB1* expression. Next, to evaluate if this effect was due to a peculiarity of 16HBE14o- cells or if it was a widespread phenomenon, we examined several cell lines. All of these showed a significant increase in *IFNB1* expression upon *MAPK15* downregulation (Figure 1D), thus confirming a generalized ability of MAPK15 to control *IFNB1* expression. Accordingly, we also performed an Enzyme-Linked Immunosorbent Assay (ELISA) to measure the amount of IFNB1 in the media of *MAPK15*-interfered cells and demonstrated that these cells secreted higher levels of this cytokine when compared to control cells (Figure 1E).

When secreted, IFNs bind to cell surface receptors and induce a signal transduction cascade, ultimately leading to increased expression of interferon-stimulated genes (ISGs). As we observed a significant difference in the amount of secreted IFNB1 in our samples, we performed RT-qPCR analysis, demonstrating the upregulation of different ISGs in cells with reduced MAPK15 levels (Figure 1F).

Polyinosinic:polycytidylic acid (poly I:C) is a synthetic analogue of double stranded RNA commonly used to induce type I IFNs based on its ability to mimic viral infections and activate innate immune responses, in particular type I interferons [25]. When using poly I:C to stimulate 16HBE14o- cells transfected with scrambled (siSCR) or MAPK15 (siMAPK15) siRNAs, we observed that the loss of MAPK15 enhanced poly I:C-induced IFNB1 response (Figure 2A). Similar results were obtained when we measured the levels of *IFNB1* gene expression by RT-qPCR (Figure 2B).

Overall, our findings therefore suggested that MAPK15 can affect *IFNB1* gene expression, secretion, and the consequent activation of its downstream genes. In particular, reduced levels of this kinase protein result in an increased *IFNB1* induction, both in basal conditions and under stimulation with poly I:C.

### 2.2. Loss of MAPK15 Impairs Nuclear Factor Kappa-Light-Chain-Enhancer of Activated B Cells (NF-kB) Activity

NF-kB binds to specific elements within the *IFNB1* promoter, acting as a key mediator in its activation and ensuring an effective production of this cytokine under conditions of cellular stress or infection [5,26]. Thus, we investigated whether the increased *IFNB1* expression caused by loss of MAPK15 could be attributed to NF-kB activation. To this aim, we used a luciferase reporter plasmid, containing five tandem NF-kB response elements, to measure NF-kB activity. We compared cells with reduced MAPK15 levels to the control, both under basal conditions (t0) and after poly I:C stimulation. Still, while unstimulated cells downregulated for *MAPK15* expression exhibited a little but significant increase in NF-kB luciferase activity (Figure 3A; t0 time point), probably attributable to cellular stress due to reduced levels of the kinase [17,27], *MAPK15* interference prevented poly I:C-dependent NF-kB activation (Figure 3A; 4 h and 6 h time points). These results suggest that the induced expression of *IFNB1* is independent of the activation of this transcription factor. Indeed, *MAPK15* overexpression enhanced NF-kB activity (Figure 3B), supporting data suggesting that this MAP kinase may act as an activator of NF-kB [28,29]. Overall, our findings therefore indicate that the increased *IFNB1* expression observed in *MAPK15*-downregulated cells is independent of NF-kB activity.

### 2.3. Reduced MAPK15 Activity Leads to JUN Activation

In addition to NF-kB, the *IFNB1* promoter contains other regulatory domains, including binding sites for transcription factors of the AP-1 family [5,26]. Thus, we next explored whether MAPK15 could affect the activity of AP-1 family members. To this end, we first assessed the AP-1 activity using a luciferase reporter plasmid containing seven AP-1 response elements. These experiments revealed a significant increase in reporter gene activity in MAPK15-silenced cells compared to control cells (Figure 4A). To gain further insight, we next tested if *MAPK15* expression could regulate, in our experimental settings, the activity of JUN, a key member of the AP-1 heterodimer involved in *IFNB1* activation. Therefore, we transfected 16HBE14o- cells with siMAPK15 or siSCR and extracted proteins at different time points. By Western blot analysis, we detected increased JUN activation, as demonstrated by its higher phosphorylation on Ser63 and Ser73 (Figure 4B), which are required for JUN to become transcriptionally active [30,31]. Indeed, samples lacking *MAPK15* expression showed an increase in the amount of active JUN over time, while its activation in control samples did not change (Figure 4B).

As the JUN protein can be phosphorylated on Ser63/73 by both JNKs and p38 MAP kinases [32], we next took advantage of their available pharmacological inhibitors to ascertain whether they are involved in controlling *IFNB1* expression downstream of MAPK15. Luciferase assays reporting *IFNB1* promoter activity revealed that the JNK inhibitor SP600125 [33] significantly reduced luciferase expression in MAPK15-silenced cells (Figure 5A). Notably, this inhibitor also markedly decreased JUN phosphorylation levels (Figure 5B), while they did not change when we inhibited p38 by using SB203580 (Figure 5C), under the same experimental conditions. Accordingly, the treatment of 16HBE14o- cells with SP600125 prevented the increase in *IFNB1* mRNA levels induced by *MAPK15* downregulation (Figure 5D).

We have previously demonstrated that MAPK15 downregulation leads to increased levels of various cytokines, including *CCL2*, *CXCL1*, *CXCL2*, *CXCL8* and *IL6* [17], suggesting a regulatory role for this kinase in controlling inflammatory responses. Accordingly, also in 16HBE14o- cells, *MAPK15* silencing resulted in a significant upregulation of several cytokines, i.e., *CCL2*, *CCL5*, *CXCL8* and *IL6*, whose expression has been already reported to be controlled by JUN [34,35,36,37,38,39] (Figure 5E). Indeed, we observed that the treatment with the JNKs inhibitor efficiently prevented the increase in the expression of all these cytokines (Figure 5E), further supporting the role of JNKs activation in driving their expression in *MAPK15*-interfered cells.

As an additional control, the same experiments using SP600125 were performed in HeLa cells, yielding similar results. Specifically, we observed reduced JUN phosphorylation and *IFNB1* expression, thereby supporting the conservation of the described molecular mechanism across cell types from different tissues (Figure 6).

Overall, our results strongly support a role for JNK-dependent JUN activation as a key event triggering *IFNB1* expression and its downstream inflammatory response induced by *MAPK15* downregulation.

### 2.4. The Antioxidant Molecule NACET Restores IFNB1 Levels

It has been well established that JNKs can be activated by reactive oxygen species (ROS) through various intracellular signaling pathways [40,41,42], ultimately promoting JUN transcriptional activation [40,43] and its role as a pro-inflammatory transcription factor. Indeed, upon ROS-dependent activation of the AP-1 complex, this readily binds to promoter regions of target genes, including *IFNB1* [44,45]. Ultimately, this mechanism allows JUN to play a central role in regulating inflammatory responses and in transmitting signals derived from oxidative stress [44,46].

Previous work in our laboratory showed that MAPK15 is a key regulator of the mitophagic process and, when its expression is reduced, cells produce high levels of mitochondria-derived ROS as a consequence of the deregulation of the important mitophagic process [12,17]. Importantly, *MAPK15* downregulation also prevents the efficient disposal of increased ROS levels by directly impairing NRF2-dependent antioxidant responses [18]. Based on this evidence, we therefore hypothesized that *MAPK15* downregulation may increase *IFNB1* expression through the oxidative stress-dependent activation of JNKs and resultant AP-1/JUN phosphorylation and activation. To test the validity of this hypothesis, we first verified that MAPK15 silencing also induced ROS accumulation in 16HBE14o- cells (Figure 7A), consistently with observations in other cell lines [12,17,18]. Once this was established, we therefore treated 16HBE14o- cells with the antioxidant molecule N-acetylcysteine ethyl ester (NACET) to counteract the effect of ROS [47]. Indeed, in these experimental conditions, we observed that NACET was able to reduce both Ser63 and Ser73 phosphorylation in a dose-dependent fashion (Figure 7B). Moreover, we also found that NACET could reduce *IFNB1* mRNA expression induced by *MAPK15* downregulation (Figure 7C), suggesting that oxidative stress was indeed the triggering event for the induction of IFNB1 levels.

## 3. Discussion

Although several aspects of the biology of the MAPK15 protein have been identified, the precise understanding of its molecular functions remain still largely incomplete to date. Here, we show that MAPK15 controls *IFNB1* gene expression, secretion, and the subsequent activation of interferon-stimulated genes such as *IFIT2*, *IFIT3* and *MX1*, as well as other pro-inflammatory cytokines. In particular, we observed that reduced levels of this MAP kinase led to an enhanced induction of *IFNB1*, under basal conditions and upon stimulation. Moreover, this phenomenon was confirmed across various cell lines, both malignant and non-malignant.

To better understand the molecular mechanisms connecting MAPK15 to *IFNB1* gene expression, we examined the activity of different transcription factors that cooperate to modulate its promoter activity. First, we explored the role of NF-kB, which is known to regulate *IFNB1* and several other genes involved in inflammatory responses. Our results showed that, in the absence of MAPK15, the NF-kB response could not be appropriately induced by pro-inflammatory stimuli. Conversely, the NF-kB activity was enhanced in cells overexpressing *MAPK15*, consistently with previous studies documenting that MAPK15 can activate NF-kB through IkBα phosphorylation [28], promote a positive feedback loop [29], and interact with the NF-kB p50 subunit [48]. We therefore excluded a role of NF-kB in the increased *IFNB1* expression caused by *MAPK15* silencing.

Next, we investigated a possible role for JUN, a member of the AP-1 family of transcription factors, in controlling IFNB1 production. We found that, upon downregulation of the MAPK15 protein, the levels of the JUN active protein are strongly increased. Specifically, we monitored JUN activation by analyzing the phosphorylation of two residues, Ser63 and Ser73 [30,31], demonstrating that this phenomenon is already evident at 24 h after silencing *MAPK15* but much more pronounced after 72 h. Interestingly, data from the literature indicate that, in gastric and colorectal cancer cells, MAPK15 can directly phosphorylate JUN on Ser63/73 [49,50]. Conversely, MAPK15 knockdown reduces levels of JUN phosphorylation [49,50], decreases total JUN protein levels, and shortens its half-life, despite the higher mRNA amount [50]. We can hypothesize that the discrepancy with our results might depend on tissue-specific mechanisms employed by gastric cells, since we observed the phenomenon of JUN phosphorylation in bronchial epithelial (16HBE14o-) and cervical adenocarcinoma cells (HeLa).

We also demonstrated that JUN activation was the result of JNKs activity, as cells’ treatment with a JNKs inhibitor, but not with a p38 MAPK inhibitor, abolished its activation. Accordingly, inhibition of JUN phosphorylation by JNKs also restored normal levels of *IFNB1* expression, indicating that the JNK-JUN axis plays a key role in increasing the IFNB1 amount in MAPK15-deficient cells. The *IFNB1* promoter contains four regulatory domains (PRD I-PRD IV), each one bound by different transcription factor complexes. The main drivers of the pathogen-induced *IFNB1* expression are IRF3 and, at a later time, IRF7 proteins. In contrast, constitutive *IFNB1* expression primarily relies on JUN and NF-kB elements [5]. Consistent with this observation, our results suggest that the absence of MAPK15 increases the levels of constitutive *IFNB1* expression, making the cells more reactive to external stimuli. In fact, cells lacking *MAPK15* expression are able to promptly respond to poly I:C, a stimulus that mimics viral infection.

By counteracting JUN activation, we also observed a reduction in the levels of the upregulated cytokines *CCL2*, *CCL5*, *IL6*, and *CXCL8.* This confirms that their expression can be modulated by the activity of this transcription factor, as already suggested by other studies [34,35,36,37,38,39,51,52].

Interestingly, our data also suggest that the oxidative stress that is generated when *MAPK15* is downregulated has a significant role in increasing cellular levels of IFNB1. Indeed, we observed that also the treatment with the antioxidant molecule NACET was able to attenuate JUN activation and the levels of *IFNB1* expression, as cells with impaired *MAPK15* expression undergo oxidative stress [17,18].

Overall, our findings indicate that reduced *MAPK15* expression induces cellular stress which, in turn, activates the transcription factor JUN, leading to elevated basal interferon levels and the induction of proinflammatory cytokines. Continuous production of type I IFNs is essential for maintaining the appropriate balance of immune activation and suppression, helping to keep the innate immune system ready to respond effectively to environmental threats. However, events that lead to hyperactivation or excessive suppression of the immune system can trigger the development of various pathologies, including immunodeficiency with recurrent infections, autoimmune diseases, chronic inflammation, and eventually tumors [3,53,54,55]. Based on our observations, MAPK15 may help mitigate proinflammatory states that could promote the development of related diseases. The protein MAPK15 plays a critical role in this regulation by influencing the production of IFNB1 JUN-driven and modulating the inflammatory response. Understanding the MAPK15 role in regulating IFNB1 and inflammation helps clarify how this pathway can be targeted to manage inflammatory and autoimmune disorders. The interplay between MAPK15 and IFNB1 provides new therapeutic opportunities to address chronic inflammatory conditions driven by dysregulated immune responses.

## 4. Materials and Methods

### 4.1. Reagents and Antibodies

The JNK-inhibitor SP600125 (Selleck Chemicals, Huston, TX, USA) was used at a final concentration of 10 or 20 μM. The p38 MAPK inhibitor SB203580 (Selleck Chemicals, Houston, TX, USA) was used at a final concentration of 10 μM. N-acetylcysteine ethyl ester (NACET) was in-house synthetized and kindly provided by Prof. Federico Galvagni. Poly I:C HMW (InvivoGen, Toulouse, France, cat.#tlrl-pic) was used at a final concentration of 5 μg/mL. The primary antibodies used were anti-MAPK1/ERK2 (Santa Cruz Biotechnology, Dallas, TX, USA sc-154), anti-phospho p38 MAPK (Thr180/Tyr182) (#9211, Cell Signaling Technology, Leiden, The Netherlands), anti-c-Jun (# 9162, Cell Signaling Technology, Leiden, The Netherlands), anti-phospho c-Jun (Ser63) (#9261, Cell Signaling Technology, Leiden, The Netherlands), and anti-phospho c-Jun (Ser73) (#9164, Cell Signaling Technology, Leiden, The Netherlands). The secondary antibody used was anti-rabbit HRP-conjugated (#111-036-003, Jackson ImmunoResearch, Ely, United Kingdom).

### 4.2. Plasmids

The used reporter plasmids were IFN-Beta_pGL3 plasmid (#102597, Addgene, Watertown, MA, USA), pNF-kB luciferase (Stratagene, San Diego, CA, USA), and pAP17X-Luc (Stratagene, San Diego, CA, USA). Other plasmid vectors used were pCEFL-HA and pCEFL-MAPK15 [56].

### 4.3. Cell Culture

The 16HBE14o- cells were maintained in a Minimum Essential Medium (MEM) supplemented with 10% fetal bovine serum (FBS), 2 mM L-glutamine, and 100 units/mL penicillin-streptomycin at 37 °C in an atmosphere of 5% CO_2_/air. HeLa cells were maintained in Dulbecco’s modified Eagle medium (DMEM) supplemented with 10% fetal bovine serum (FBS), 2 mM L-glutamine, and 100 units/mL penicillin-streptomycin at 37 °C in an atmosphere of 5% CO_2_/air.

### 4.4. MAPK15 Silencing

MAPK15-specific siRNA (target sequence 5′-TTGCTTGGAGGCTACTCCCAA-3′) and non-silencing scramble siRNA (target sequence 5′-AATTCTCCGAACGTGTCACGT-3′) were obtained from QIAGEN (QIAGEN, Milan, Italy). siRNAs were transfected at a final concentration of 100 nM using the Lipofectamine RNAiMAX Reagent (Thermo Fisher Scientific, Monza (MB), Italy). Samples were collected at 24, 48, and 72 h after transfection.

### 4.5. Luciferase Assays

In all experiments, 1 μg of total plasmid vectors was transfected using Lipofectamine LTX (Thermo Fisher Scientific, Monza (MB), Italy), according to manufacturer’s instructions. In silencing experiments, cells were seeded in 6-well cell culture plates and, 24 h later, subjected to transfection with scramble or MAPK15 siRNAs. After 24 h, cells were transfected with the reporter plasmid. In overexpression experiments, cells were seeded in 6-well cell culture plates and, 24 h later, co-transfected with pCEFL-HA empty vector or with pCEFL-MAPK15 plasmid, together with the reporter plasmid. Twenty-four or forty-eight hours after plasmid transfection, cells were lysed in a Passive Lysis Buffer (Promega, Milan, Italy) and the luciferase activity was assessed by a GloMax 20/20 Luminometer (Promega, Milan, Italy), using the Luciferase Assay System (Promega, Milan, Italy).

### 4.6. RT-qPCR

Total RNA was extracted using the QIAzol Lysis Reagent (QIAGEN, Milan, Italy). Additionally, 1 μg RNA was reverse transcribed using the QuantiTect Reverse Transcription Kit (QIAGEN, Milan, Italy). qPCR was performed by the Luna Universal qPCR Master Mix (New England Biolabs, Ipswich, MA, USA) in a Rotor-Gene 6000 Real-Time PCR detection system (Corbett Life Science, Sydney, Australia). Beta-2-Microglobulin (*B2M*) was used as housekeeper gene. The primers used are shown in Appendix A.

### 4.7. ELISA Assay

To quantify the amount of secreted IFN-beta, 8 × 10^4^ 16HBE14o- cells were seeded in 6-well plates and, 24 h later, were transfected with scramble or MAPK15 siRNA. Forty-eight hours after transfection, the supernatants of four wells for each condition were pooled. Freshly collected samples were concentrated using Amicon Ultra-4 centrifugal filter devices with MWCO 3KDa (Merck Millipore, Darmstadt, Germany). Concentrated samples of about 500 μL were immediately stored at −80 °C and analyzed within a few days (<5). The amount of IFN-beta in the samples was assessed by the VeriKine Human Interferon-Beta ELISA Kit (PBL Assay Science, Piscataway, NJ, USA), following the manufacturer’s instructions. The absorbance at 450 nm was measured by a VersaMax microplate reader (Molecular Devices, San Jose, CA, USA).

### 4.8. Western Blots

Total lysates were obtained by resuspending cellular pellets in a RIPA buffer (50 mM TRIS-HCl pH 8.0, 150 mM NaCl, 0.5% sodium deoxycholate, 0.1% SDS, 1% NP-40) with the addition of protease inhibitors (cOmplete Protease Inhibitor cocktail, EDTA-free; Roche Italia, Monza (MB), Italy) and phosphatase inhibitors (2 mM NaF, 2 mM Na_3_VO_4_; Sigma Aldrich, Milan, Italy). Total proteins were quantified by the Bradford assay and the same quantity of lysates was used for Western blot analysis. The Laemmli Loading Buffer 5X (250 mM Tris-HCl pH 6.8, 10% SDS, 50% glycerol, bromophenol blue) was added to the protein samples, which were then heated for 5 min at 95 °C. Lysates were loaded on SDS-PAGE poly-acrylamide gel, transferred to an Immobilon-P PVDF membrane (Merck Millipore, Darmstadt, Germany), probed with the appropriate antibodies, and revealed by enhanced chemiluminescence detection (ECL Plus; GE Healthcare, Milan, Italy). Densitometric analysis of Western blots was performed with NIH Image J Version 1.54p (National Institutes of Health, Bethesda, MD, USA).

### 4.9. ROS Measurement

Intracellular ROS levels were assessed using the ROS-sensitive probe CM-H2DCFDA (C6827, Invitrogen, Monza (MB), Italy). Cells were incubated for 30 min with 1 μM CM-H2DCFDA in a serum-free medium, and fluorescence was measured using a NovoCyte Quanteon Flow Cytometer (Agilent Technologies, Cernusco sul Naviglio, Milan, Italy). Data analysis was performed with the FlowJo software v10.10 (BD Biosciences, Franklin Lakes, NJ, USA).

### 4.10. Statistics

The results are presented as mean ± SD of at least three independent experiments. Significance was assessed by the unpaired *t*-test with Welch’s correction, using the GraphPad Prism6 software. Asterisks were attributed as follows: * *p* < 0.05, ** *p* < 0.01, *** *p* < 0.001, **** *p* < 0.0001.

## Figures and Tables

**Figure 1 ijms-26-05148-f001:**
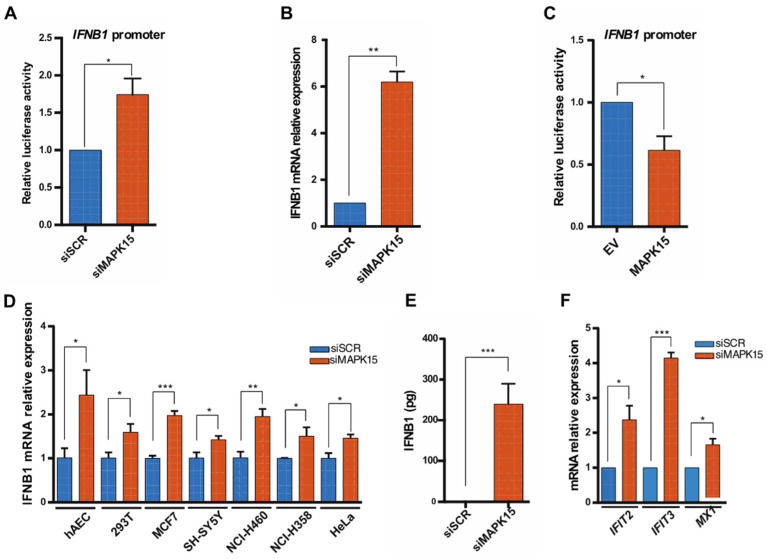
Reduced MAPK15 levels raise IFNB1 expression. (**A**) *IFNB1* promoter reporter assay in human 16HBE14o- bronchial epithelial cells, with downregulated *MAPK15*. Cells were transfected with siRNA against MAPK15 (siMAPK15) or control scramble siRNA (siSCR) for 72 h. (**B**) RT-qPCR analysis of *IFNB1* mRNA expression in 16HBE14o- cells silenced as in (**A**). (**C**) *IFNB1* promoter reporter assay in human 16HBE14o-, with overexpressed *MAPK15.* Cells were transfected with the vector containing the *MAPK15* gene (MAPK15) or the control empty vector (EV) for 24 h. (**D**) RT-qPCR analysis of *IFNB1* mRNA expression in different human cell lines silenced as in (**A**). (**E**) ELISA detection of IFNB1 in media of 16HBE14o- cells, after 48 h of silencing with scramble or MAPK15 siRNA. (**F**) RT-qPCR analysis of some interferon-stimulated genes in 16HBE14o- cells transfected as in (**A**). * *p* < 0.05, ** *p* < 0.01, *** *p* < 0.001.

**Figure 2 ijms-26-05148-f002:**
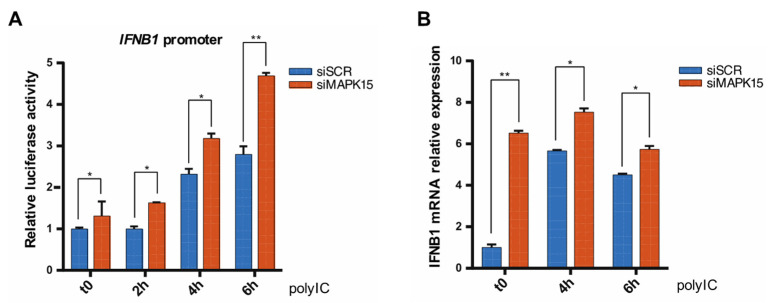
Cells lacking MAPK15 express higher levels of *IFNB1* under stimulation. (**A**) *IFNB1* promoter reporter assay in human bronchial epithelial cells 16HBE14o-, transfected with control scramble siRNA (siSCR) or siRNA against MAPK15 (siMAPK15) for 72 h, and treated with 5 μg/mL poly I:C for up to 6 h. (**B**) RT-qPCR analysis of *IFNB1* mRNA expression in 16HBE14o- cells transfected and treated as in (**A**). * *p* < 0.05, ** *p* < 0.01.

**Figure 3 ijms-26-05148-f003:**
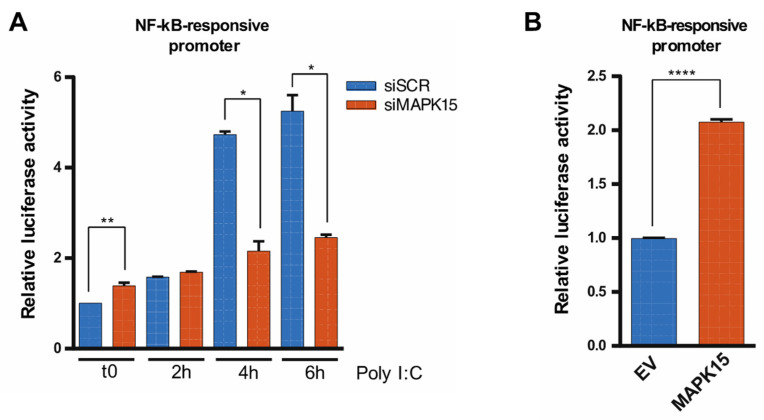
MAPK15 expression affects NF-kB activity. (**A**) NF-kB-responsive promoter reporter assay in human 16HBE14o- bronchial epithelial cells, transfected with control scramble siRNA (siSCR) or siRNA against MAPK15 (siMAPK15) for 72 h, and treated with 5 μg/mL poly I:C for up to 6 h. (**B**) NF-kB promoter reporter assay in human bronchial epithelial cells 16HBE14o-, transfected with control empty vector (EV) or MAPK15 expressing vector (MAPK15) for 24 h. * *p* < 0.05, ** *p* < 0.01, **** *p* < 0.0001.

**Figure 4 ijms-26-05148-f004:**
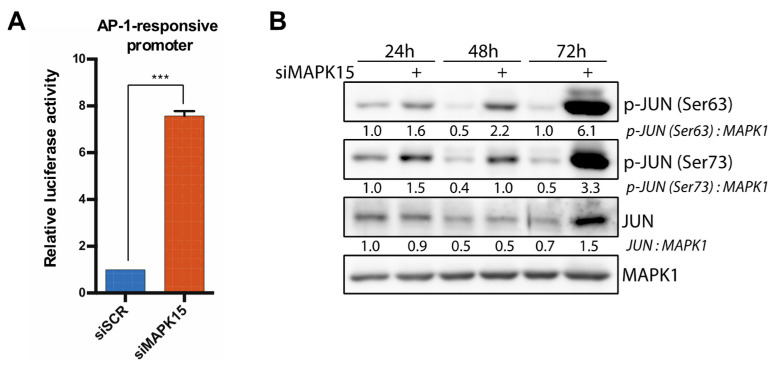
Reduced *MAPK15* expression induces AP-1 and JUN activation. (**A**) AP-1-responsive promoter reporter assay in human bronchial epithelial cells 16HBE14o-, transfected with control scramble siRNA (siSCR) or siRNA against MAPK15 (siMAPK15) for 72 h. (**B**) Western blot analysis showing levels of JUN phosphorylation in 16HBE14o- cells transfected with control scramble siRNA (siSCR) or siRNA against *MAPK15* (siMAPK15), at different time points. *** *p* < 0.001.

**Figure 5 ijms-26-05148-f005:**
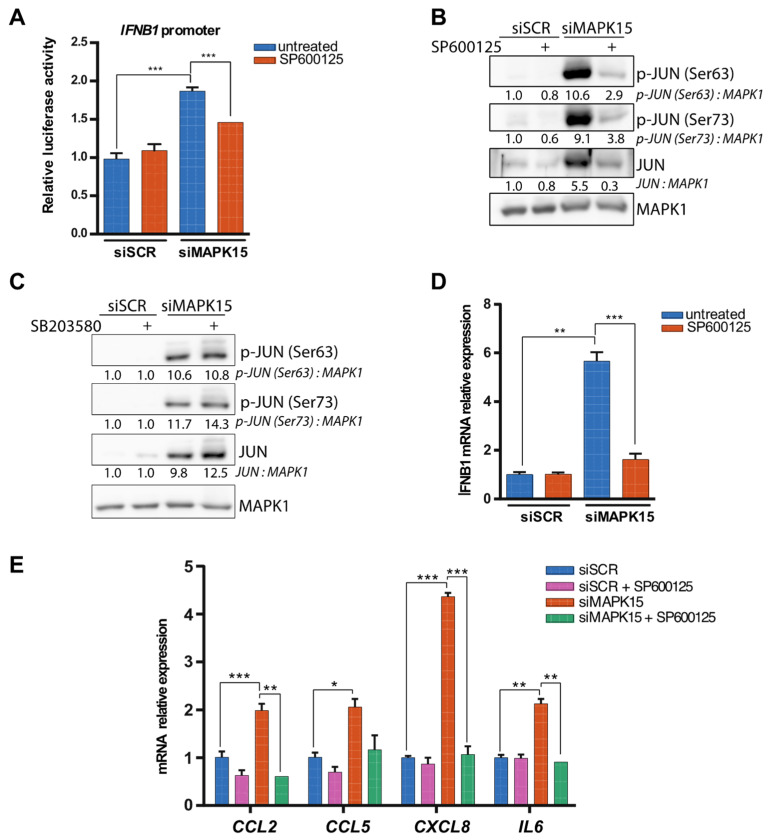
Inhibition of c-Jun N-terminal kinase reduces JUN activation and *IFNB1* expression. (**A**) *IFNB1* promoter reporter assay in human 16HBE14o- bronchial epithelial cells, treated with control scramble siRNA (siSCR) or siRNA against *MAPK15* (siMAPK15) for 72 h, and with 10 μM c-Jun N-terminal kinase inhibitor SP600125 for 24 h (**B**). Western blot analysis showing levels of JUN phosphorylation in 16HBE14o- cells, transfected with control scramble siRNA or siRNA against *MAPK15*, for 72 h, and treated with 20 μM SP600125 for 24 h. (**C**) Western blot analysis showing levels of JUN phosphorylation in 16HBE14o- cells, transfected with control scramble siRNA or siRNA against MAPK15, for 72 h, and treated with 10 μM p38 MAPK inhibitor SB203580 for 24 h. (**D**) RT-qPCR analysis of *IFNB1* mRNA expression in 16HBE14o- cells treated as in (**B**). (**E**) Analysis of cytokine expression in 16HBE14o- cells treated as in (**B**). * *p* < 0.05, ** *p* < 0.01, *** *p* < 0.001.

**Figure 6 ijms-26-05148-f006:**
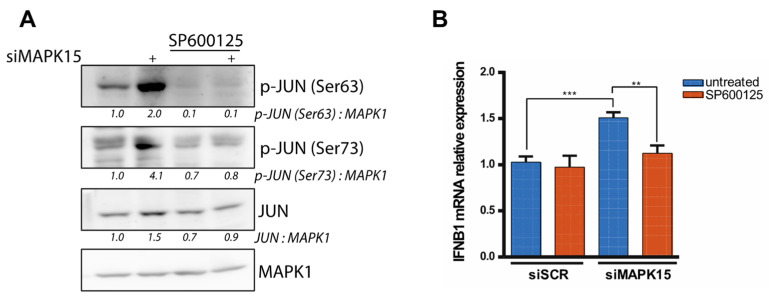
Inhibition of c-Jun N-terminal kinase reduces JUN activation and *IFNB1* expression in *MAPK15*-silenced HeLa cells. (**A**) Western blot analysis showing levels of JUN phosphorylation in HeLa cells, transfected with control scramble siRNA (siSCR) or siRNA against MAPK15 (siMAPK15) for 48 h, and treated with 20 μM c-Jun N-terminal kinase inhibitor SP600125 for 24 h. (**B**) RT-qPCR analysis of *IFNB1* mRNA expression in HeLa cells treated as in (A). ** *p* < 0.01, *** *p* < 0.001.

**Figure 7 ijms-26-05148-f007:**
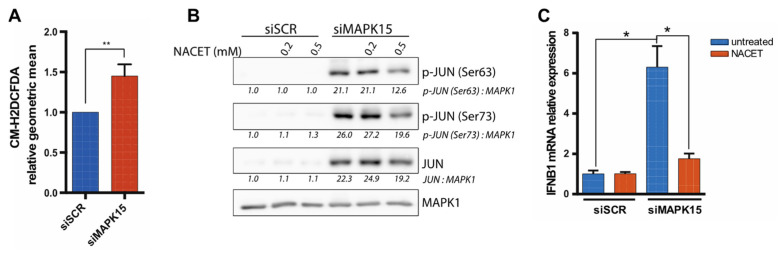
Oxidative stress induces JUN-dependent activation of *IFNB1* expression. (**A)** ROS levels measured by flow cytometry using the CM-H2DCFDA probe in 16HBE14o- cells, transfected with control scramble siRNA (siSCR) or siRNA against MAPK15 (siMAPK15), for 72 h. (**B**) Western blot analysis showing levels of JUN phosphorylation in 16HBE14o- cells, transfected as in (A) and treated with 0.2 or 0.5 mM NACET, for 24 h. (**C**) RT-qPCR analysis of *IFNB1* mRNA expression in 16HBE14o- cells transfected as in (A) and treated with 0.5 mM NACET for 24 h. * *p* < 0.05, ** *p* < 0.01.

## Data Availability

The raw data supporting the conclusions of this articles will be made available by the authors on request.

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
