# Peer review of "MAPK15 Prevents IFNB1 Expression by Suppressing Oxidative Stress-Dependent Activation of the JNK-JUN Pathway"

_ijms, 2025, doi:10.3390/ijms26115148_

Round 1

Reviewer 1 Report

Comments and Suggestions for Authors

n this manuscript, Monia Taranta and colleagues identify MAPK15 as a modulator of IFNB1 expression and propose that pharmacological targeting of MAPK15 may represent a novel therapeutic strategy for chronic inflammatory conditions characterized by dysregulated cytokine production. Overall, the study is interesting and potentially impactful; however, several important issues should be addressed:

Abstract: The authors should briefly introduce IFNB1 in the background section of the abstract, including its biological significance. In addition, they should clearly state the potential relevance of discovering a novel mechanism involving MAPK15-mediated regulation of IFNB1 production.

Figure 1A: Why was a mutant form of IFNB1 not designed or included in the experimental setup? Including a mutation could strengthen the functional interpretation.

Luciferase reporter assays: For all luciferase reporter experiments, it is important to describe the promoter sequences used, especially potential transcription factor binding sites. This critical information is missing. Furthermore, the inclusion of promoter mutant constructs would be necessary to confirm specific binding and regulatory effects.

Figure 5: The authors hypothesize that MAPK15 downregulation may enhance IFNB1 expression through oxidative stress-dependent mechanisms. However, no direct evidence of oxidative stress is provided. Experiments such as ROS staining or other oxidative stress assays should be performed to validate this hypothesis.

Western blots: Throughout the manuscript, Western blot (WB) analyses lack appropriate loading controls. Moreover, in Figure 4A, basal expression of p-JUN and JUN is shown, but in other experimental conditions these proteins appear absent or undetectable. The authors should clarify these inconsistencies and include proper normalization to internal controls.

Author Response

Comments and Suggestions for Authors.

Comment 1: In this manuscript, Monia Taranta and colleagues identify MAPK15 as a modulator of IFNB1 expression and propose that pharmacological targeting of MAPK15 may represent a novel therapeutic strategy for chronic inflammatory conditions characterized by dysregulated cytokine production. Overall, the study is interesting and potentially impactful; however, several important issues should be addressed:

Response 1: We thank the Reviewer for her/his appreciation, and we believe we addressed all raised points, as detailed below.

Comment 2: Abstract: The authors should briefly introduce IFNB1 in the background section of the abstract, including its biological significance. In addition, they should clearly state the potential relevance of discovering a novel mechanism involving MAPK15-mediated regulation of IFNB1 production.

Response 2: DONE. We revised the abstract to briefly introduce IFNB1, its biological relevance and the health implications of its dysregulation (please, see page 1, lines 13-15). We also emphasize the significance of identifying a novel regulatory mechanism—such as the one mediated be MAPK15 described in this study—particularly in the context of chronic inflammatory diseases (please, see page 1, lines 25-29). We thank the reviewer for this suggestion. 

Comment 3: Figure 1A: Why was a mutant form of IFNB1 not designed or included in the experimental setup? Including a mutation could strengthen the functional interpretation.

Response 3: We thank the reviewer for this observation. Unfortunately, as now depicted in the new Suppl. Fig.1, (please, see also below in the next response to Reviewer 1), the IFNB1 promoter is very complex as composed to elements responsive to several transcription factors (i.e., NF-kB, AP-1 and IRFs) which are contained in the short sequence used in the reporter plasmid used in this manuscript. Consequently, as we do not currently own a mutated form of the promoter in each of its already reported responsive elements, we should sequentially mutagenize it in each site and next combine them in every arrangement of WT and mutated elements, which would surely take more time than what granted to us from the Journal. Still, as an alternative approach to respond to the reviewer’s observation, we have now conducted additional luciferase assays as described below: i) in new Fig. 5A, to functionally assess the involvement of JUN in the regulation of the IFNB1 promoter reporter, we examined the effects of the JNKs inhibitor SP600125 on the activity of the reporter plasmid, upon downregulation of MAPK15 expression (please, see also page 5, lines 182-184); ii), to confirm a role for MAPK15 in specifically controlling the activity of AP-1/JUN transactivating function, we also now included, in the new Fig. 4A, an assay performed on an AP-1 luciferase reporter plasmid (containing only AP-1 responsive elements) to evaluate the specificity of the approach under our experimental conditions (please, see also page 5, lines 159-162).

In this context, we also want to underline that our analysis is not limited to the artificial IFNB1 promoter reporter plasmid but also confirmed on the complete endogenous promoter, by extensive RT-qPCR assays. We therefore hope our approaches and further experiments contained in this revised manuscript will be acceptable for the Reviewer, based on the consistency across these independent methods supporting the reliability and robustness of our findings.

Comment 4: Luciferase reporter assays: For all luciferase reporter experiments, it is important to describe the promoter sequences used, especially potential transcription factor binding sites. This critical information is missing. Furthermore, the inclusion of promoter mutant constructs would be necessary to confirm specific binding and regulatory effects.

Response 4: DONE. In response to the reviewer’s suggestion, we have now included detailed information in the text regarding the regulatory elements present in the plasmids used for the luciferase assays (please, see page 2, lines 81-82; page 4, line 138). To enhance clarity, we have also included a new supplementary figure (now Figure S1) that illustrates the specific regulatory regions contained in the IFN-β_pGL3 plasmid.

Regarding the second observation, please see our response to the previous point raised from the Reviewer.

Comment 5: Figure 5: The authors hypothesize that MAPK15 downregulation may enhance IFNB1 expression through oxidative stress-dependent mechanisms. However, no direct evidence of oxidative stress is provided. Experiments such as ROS staining or other oxidative stress assays should be performed to validate this hypothesis.

Response 5: DONE. Previous works from our laboratory repeatedly demonstrated that reduced MAPK15 expression leads to elevated levels of ROS in different cellular systems and experimental conditions (Rossi et al., Oncotarget, 2016; Franci et al., Aging Cell, 2022; Franci et al., Redox Biology, 2024). However, since those findings were obtained in different cell lines, we also agree with the Reviewer that we need to formally prove that this mechanism is maintained also in the specific cell line used in this manuscript: we therefore now conducted new experiments to measure ROS levels in 16HBE14o- cells upon MAPK15 downregulation (please, see new Fig. 7A and page 7, lines 240-241), ultimately validating our experimental hypothesis in the currently used model.

Comment 6: Western blots: Throughout the manuscript, Western blot (WB) analyses lack appropriate loading controls. Moreover, in Figure 4A, basal expression of p-JUN and JUN is shown, but in other experimental conditions these proteins appear absent or undetectable. The authors should clarify these inconsistencies and include proper normalization to internal controls.

Response 6: We have long-term experience in the study of MAP kinases and have noticed that MAPK1 (ERK2) protein and mRNA levels are very poorly influenced (if at all) by the most different stimuli we have tested. Especially, we have noticed that actin or other cytoskeletal proteins are much more influenced by stimuli that comparably do not affect ERK2 expression. Importantly, as Guidelines for the use and interpretation of assays for monitoring autophagy (Klionsky et al., Autophagy, 2012) specifically warn about the possibility that actin levels are very much influenced by changes in the cellular autophagic activity, we chose not to use actin as internal normalization control for our experiments as we demonstrated that MAPK15 is a powerful regulator of autophagy (see, for example, Colecchia et al., Autophagy, 2012; Colecchia et al., Autophagy, 2015; Colecchia et al., JBC, 2018). Based on this, we long ago decided to use ERK2 as loading control for our WB experiments. Furthermore, we have also previously performed confirmatory experiments (please see Colecchia et al., Autophagy 2012) to compare MAPK1/ERK2 with GAPDH and/or actin protein levels in response to different treatments, confirming that ERK2 can be successfully used for normalization purposes. We therefore respectfully ask to maintain the use of MAPK1/ERK2 as a loading control for our experiments in this manuscript.

Regarding the differences in the levels of the JUN protein and of its phosphorylated levels in old Fig. 4, we believe that they depend on the different times of exposure of the blots in the independent experiments. As conditions and treatments change in the different experiments (for example by using different drugs for different times to inhibit p38 or JNKs, possibly associated to siRNA treatments), it is difficult to obtain absolute levels of the different proteins perfectly identical in the different conditions tested. Nonetheless, we feel that the constant relative differences among protein levels in the different experiments supports our experimental conclusions and the ability of MAPK15 to control the activity of the JUN transcription factor in a JNK-dependent fashion. We thank the reviewer for the comment which helped us to clarify these issues.

We are especially grateful for the efforts of the two Reviewers and believe that they have truly helped us to strengthen the manuscript. We are appreciative of your willingness to consider the paper for publication in the International Journal of Molecular Sciences.

Reviewer 2 Report

Comments and Suggestions for Authors
  1. Mitogen activated protein kinases (MAPKs or MAP kinases) are signaling proteins that efficiently transduce and amplify extracellular and intracellular stimuli, inducing re sponses that affect several cellular functions such as gene expression, proliferation, me- 47 tabolism, motility, survival, apoptosis and differentiation. Many recent studies have https://doi.org/10.1371/journal.pone.0311954 highlighted importance of MAPK in cancer therapeutics. These must be cited at relevant places and also the novelty of the study needs to be highlighted at the end of Introduction section of your manuscript.
  2. I feel the Introduction section lacks a base and connectivity. A brief idea about kinases and their importance in cancer therapeutics should be given in 2 3 lines.
  3. To this aim, we decided to use the bronchial epithelial cell line 16HBE14o- since the bronchial epithelium, which expresses high levels of endogenous MAPK15 protein,. Is this the highest expressing cell line for MAPK15 protein? HEK293 and certain colon carcinoma cell lines (e.g., HCT116) have also been reported in literature to express moderate to high levels of MAPK15. Why these were eliminated?
  4. Figures from supplementary data could be moved to main text.
  5. Overall the work is nice backed up sound methodology and results are presented in a good way, I recommend acceptance after addressing few minor comments.
  6. English language needs to be checked, long sentences must be broken to small and proofread for any grammatical errors.

Author Response

Comments and Suggestions for Authors

Comment 1: Mitogen activated protein kinases (MAPKs or MAP kinases) are signaling proteins that efficiently transduce and amplify extracellular and intracellular stimuli, inducing responses that affect several cellular functions such as gene expression, proliferation, metabolism, motility, survival, apoptosis and differentiation. Many recent studies have highlighted importance of MAPK in cancer therapeutics (https://doi.org/10.1371/journal.pone.0311954). These must be cited at relevant places and also the novelty of the study needs to be highlighted at the end of Introduction section of your manuscript.

Response 1: DONE. In the Introduction section, we have now further highlighted the relevance of MAPK signaling in cancer therapeutics and have included the additional reference (please, see page 2, line 55). Additionally, we added a sentence at the end of the specific section to emphasize the potential therapeutic implications of our findings and better contextualize the significance of this study (please, see page 2, line 69). We thank the reviewer for her/his valuable suggestions.

Comment 2: I feel the Introduction section lacks a base and connectivity. A brief idea about kinases and their importance in cancer therapeutics should be given in 2 3 lines.

Response 2: DONE. In the Introduction section, we added a few sentences to better clarify the significance of MAPKs, which have emerged as attractive targets, particularly in the context of cancer therapeutics (please, see page 2, lines 51-56).

Comment 3: To this aim, we decided to use the bronchial epithelial cell line 16HBE14o- since the bronchial epithelium, which expresses high levels of endogenous MAPK15 protein. Is this the highest expressing cell line for MAPK15 protein? HEK293 and certain colon carcinoma cell lines (e.g., HCT116) have also been reported in literature to express moderate to high levels of MAPK15. Why these were eliminated?

Response 3: Based on high MAPK15 expression in human lungs and on previous reports involving this kinase in chronic obstructive pulmonary disease (COPD), we recently strongly focused the attention of our laboratory on lung physiology, with specific attention to a potential role for MAPK15 in controlling oxidative stress in this tissue. Indeed, because of their function, lungs and their epithelial cells are highly exposed to ROS mediated damage, both because of their endogenous production, which is facilitated by high oxygen concentrations in inhaled air, but also because they are continuously exposed to external insults such as air pollutants and cigarette smoke. Accordingly, lungs have developed very effective systems of antioxidants located both inside and outside the cells, to cope with detrimental effects due to continuous oxidative stress. Importantly, when external insults overcome the antioxidant capacity of the organ, either due to their high doses or prolonged duration, several lung diseases, such as asthma, COPD, fibrosis, and cancer may result. There is, therefore, huge interest in understanding molecular mechanisms controlling its activity, to develop innovative pharmacological approaches against different diseases having oxidative stress and inflammation as underlying pathological features.

In this context, we recently demonstrated that MAPK15 is able control multiple steps involved in preventing intracellular damages due to excessive ROS accumulation, making this kinase an almost ideal target for preventing physio-pathological consequences of oxidative stress, i.e., inflammation, aging and cancer. Indeed, we showed that MAPK15 prevents ROS accumulation by controlling the mitophagic process (Franci et al., Aging Cell, 2022) and to enhance NRF2-dependent responses, removing ROS once they are formed (Franci et al., Redox Biology, 2024).

Importantly, we also demonstrated that MAPK15 prevents cellular senescence in lung cells, including the secretion of different cytokines (i.e., CCL2, CXCL1, CXCL2, IL6 and IL8) participating to the senescent-associated secretory phenotype (SASP) of these cells, suggesting that this kinase may also really coordinate inflammatory responses in the lung, which support and/or are responsible for the most important and frequent deadly diseases of this tissue, namely, fibrosis, COPD and cancer.

The submitted manuscript is therefore included in our recent research interest in understanding the molecular bases of lung physiology and pathology and, for this reason, we concentrated our interest on a model of immortalized but non-transformed bronchial cells, 16HBE14o- cells, so that they may represent a significant model for normal bronchial cells, which serve as a primary barrier to external agents and plays a key role in host defense. As expected, they express appreciable levels of MAPK15 mRNA and protein, as we previously confirmed (in the cited references) in several other bronchial and lung normal and transformed cells. Nonetheless, as noticed by the Reviewer, we initially confirmed that the general mechanism we propose, i.e. the ability of MAPK15 to control IFNB1 expression, is not specific for bronchial cells because it is maintained in cells originating from different tissues (kidney, breast, ovary, nervous system) (see Fig. 1D). Then, after this demonstration, we focused again to our most significant model, represented by bronchial cells and just omitted to repeat each single experiment in each cell line, to avoid unnecessary repetition of the same result. We are nonetheless available to repeat specific experiments in specific cells in case the Reviewer requires for specific aims. We hope the Reviewer will find our approach now acceptable and thank him to allow us to better explain our vision on the general subject.

Comment 4: Figures from supplementary data could be moved to main text.

Response 4: As suggested by the reviewer, we moved Figure S2 in the main text, where it now appears as Figure 6.

Comment 5: Overall, the work is nice backed up sound methodology and results are presented in a good way, I recommend acceptance after addressing few minor comments.

Response 5: We thank the reviewer for such positive feedback.

Comment 6: English language needs to be checked, long sentences must be broken to small and proofread for any grammatical errors.

Response 6: We have now tried to more carefully revise the manuscript to improve the clarity and readability of the text. We have broken down long sentences into shorter, more concise ones and thoroughly proofread the manuscript to correct any grammatical errors.

We are especially grateful for the efforts of the two Reviewers and believe that they have truly helped us to strengthen the manuscript. We are appreciative of your willingness to consider the paper for publication in the International Journal of Molecular Sciences.

Round 2

Reviewer 1 Report

Comments and Suggestions for Authors

The authors have addressed all previous concerns, and the manuscript is now suitable for publication.